# HepatoPredict Accurately Selects Hepatocellular Carcinoma Patients for Liver Transplantation Regardless of Tumor Heterogeneity

**DOI:** 10.3390/cancers17030500

**Published:** 2025-02-02

**Authors:** Rita Andrade, Judith Perez-Rojas, Sílvia Gomes da Silva, Migla Miskinyte, Margarida C. Quaresma, Laura P. Frazão, Carolina Peixoto, Almudena Cubells, Eva M. Montalvá, António Figueiredo, Augusta Cipriano, Maria Gonçalves-Reis, Daniela Proença, André Folgado, José B. Pereira-Leal, Rui Caetano Oliveira, Hugo Pinto-Marques, José Guilherme Tralhão, Marina Berenguer, Joana Cardoso

**Affiliations:** 1Surgery Department, Centro Hospitalar e Universitário de Coimbra, 3004-561 Coimbra, Portugal; 11831@ulscoimbra.min-saude.pt (R.A.); jgtralhao@ulscoimbra.min-saude.pt (J.G.T.); 2Faculty of Medicine, University of Coimbra, 3004-504 Coimbra, Portugal; rui.caetano@germanodesousa.com; 3Pathology Service, Hospital Universitari i Politècnic La Fe, 46026 Valencia, Spain; perez_jud@gva.es; 4Centro de Investigación Biomédica en Red de Enfermedades Hepáticas y Digestivas (CIBERehd), Instituto de Salud Carlos III, 28029 Madrid, Spain; almudena_cubells@iislafe.es (A.C.); montalva_eva@gva.es (E.M.M.); marina.berenguer@uv.es (M.B.); 5Instituto de Investigación Sanitaria La Fe (ISS La Fe), 46026 Valencia, Spain; 6Hepato-Biliary-Pancreatic and Transplantation Centre, Hospital Curry Cabral, Unidade Local de Saúde de São José, 1069-166 Lisbon, Portugal; silvia.silva@ulssjose.min-saude.pt (S.G.d.S.);; 7NOVA Medical School, 1169-056 Lisbon, Portugal; 8Ophiomics Precision Medicine, 1600-514 Lisbon, Portugal; mmiskinyte@ophiomics.com (M.M.); mquaresma@ophiomics.com (M.C.Q.); laurapmffrazao@gmail.com (L.P.F.); anacpeixoto@tecnico.ulisboa.pt (C.P.); mreis@ophiomics.com (M.G.-R.); dproenca@ophiomics.com (D.P.); afolgado@ophiomics.com (A.F.); jleal@ophiomics.com (J.B.P.-L.); 9Hepatology Unit, Hospital Universitari i Politècnic La Fe, 46026 Valencia, Spain; 10Liver Transplantation and Surgery Unit, Hospital Universitari I Politècnic La Fe, 46026 Valencia, Spain; 11Facultad de Medicina, Universidad de Valencia, 46010 Valencia, Spain; 12Pathology Service, Hospital Curry Cabral, Unidade Local de Saúde de São José, 1069-166 Lisbon, Portugal; antonio.figueiredo@ulssjose.min-saude.pt; 13Pathology Department, Unidade Local de Saúde de Coimbra, 3004-561 Coimbra, Portugal; macipriano@ulscoimbra.min-saude.pt; 14Coimbra Institute for Clinical and Biomedical Research (iCBR), 3000-548 Coimbra, Portugal; 15Centro de Investigação em Meio Ambiente, Genética e Oncobiologia (CIMAGO), 3001-301 Coimbra, Portugal; 16Centro Académico e Clínico (CAC), 3004-531 Coimbra, Portugal

**Keywords:** HepatoPredict, HCC, tumor heterogeneity, liver transplant, multi-target genomic assay, prognostic test, liver biopsy

## Abstract

Liver cancer is a leading cause of death and liver transplants (LT) offer the best chance of survival for many patients. However, current methods to decide who should receive a transplant often fall short, leaving some patients without access to life-saving care. This research focuses on improving these decisions with HepatoPredict (HP). This new tool uses technology to combine tumor and patient data to predict how well someone will do after a transplant. HP was tested on a large group of patients and proved to be more accurate than current methods. It also performed well even when samples came from different parts of the tumor. These findings could help medical teams make fairer and more reliable transplant decisions, ultimately improving patient outcomes and advancing how liver cancer is managed in the medical community.

## 1. Introduction

Hepatocellular carcinoma (HCC), the most common form of primary liver cancer (78–85%) [1], is a leading cause of cancer-related death worldwide [2]. Although liver cancer incidence has declined in some regions since 2000, such as Southeast Asia, East Asia and Oceania and sub-Saharan Africa, it has increased in others, including Central and Eastern Europe, Central Asia, Latin America and the Caribbean, North Africa and the Middle East [3]. The global annual number of new liver cancer cases is predicted to rise by 55% between 2020 and 2040 [4]. As a result, HCC poses a substantial and growing economic burden to healthcare systems globally [5]. Additionally, the symptoms and complications of HCC, particularly in advanced stages, have a profound negative impact on the physical, emotional and functional well-being of patients [5].

Over the past decades, significant advances have been made in non-surgical treatments for HCC, including chemoembolization, ablation, radiation and systemic therapies, which now provide a range of alternatives to patients. However, liver resection and liver transplantation (LT) remain the main curative-intent options for early to intermediate HCC [6]. Despite the success of liver resection surgeries, these are still associated with lower survival rates and higher recurrence rates [7], prompting LT as the best treatment for HCC.

Due to the scarcity of liver donors and the significant economic and social burden associated with LT, several criteria are used globally to select patients for this procedure, prioritizing those with a lower risk of recurrence. Most of the standard criteria are morphological, relying solely on imaging techniques, which lack sensitivity [8,9,10,11,12,13,14,15,16,17] and do not truly account for the influence of tumor biology on patient outcomes. Even criteria that incorporate biological markers as a proxy for tumor behavior (e.g., alpha-fetoprotein, AFP, or des-gamma carboxyprothrombin, DCP, also known as protein-induced by vitamin K absence/antagonist-II, PIVKA-II) have limited sensitivity and specificity. Elevated AFP levels have been reported in chronic liver diseases, particularly cirrhosis, even in the absence of HCC [18]. Moreover, AFP and DCP levels can fluctuate based on patient characteristics [19] and AFP expression can be influenced by tumor size [20]. Validation of the clinical utility of these biomarkers by larger multicentric long-term phase III studies is required [21]. As a result, most HCC management guidelines do not recommend these biomarkers for routine screening or consider them optional [22].

The strict Milan criteria, a cornerstone in patient selection for LT, ensure that less than 20% of selected patients will experience recurrence [23]. However, this criterion also excludes many patients who could benefit from surgery. Several extended criteria are used clinically to address this limitation, including both morphological criteria (University of California, San Francisco—UCSF, up-to-seven) and those incorporating tumor biology surrogate AFP (AFP score, MetroTicket 2.0). These criteria aim to expand access to LT while still achieving favorable outcomes [23,24]. Nonetheless, the use of extended criteria also leads to an increase in false positives—patients who ultimately recur after LT [23]. The inability to accurately select LT candidates remains a significant limitation of the existing tools, rendering them suboptimal. This issue raises ethical concerns regarding equity, as it includes patients with poor prognosis who are unlikely to benefit from LT, while wrongly excluding patients with good prognosis who could benefit from the procedure [23,25,26].

The HepatoPredict tool (HP) was developed in response to this unmet need and has undergone continuous refinement and analytical validation [27,28]. HP uses a proprietary machine learning model that integrates molecular and clinical features. Specifically, gene expression data from a tumor sample obtained through a needle biopsy is combined with clinical variables (number of tumors and diameter of the largest tumor and total tumor diameter). This enables the classification of patients based on their predicted capacity to benefit from LT. Patients are categorized into those likely to benefit from LT (with high confidence and a subset of these with very high confidence) and those with no predicted benefit [27].

It has been previously shown that the HP tool outperforms existing clinical criteria in identifying HCC patients who are most likely to benefit from LT. To our knowledge, HP is the only LT selection tool that directly assesses tumor biology [27,28]. The performance of this machine learning-based tool improves when trained with representative and increasingly larger datasets from the relevant population [28]. Nonetheless, there has been a lack of data regarding the sampling bias introduced by the biopsy procedure, particularly in the context of HCC heterogeneity.

Tumor heterogeneity is commonly observed in many malignancies, including HCC, which is characterized by multiple layers of heterogeneity [29]. These variations can occur in three main contexts: within a single tumor (intra-tumor heterogeneity), across independent tumor sites in the same patient (intra-patient) and between tumors of different individuals (inter-patient heterogeneity) [30,31]. Intra-tumor and intra-patient heterogeneity lead to subclonal populations of tumor cells, which may develop due to genetic or epigenetic changes. These subclones gain a fitness advantage in specific contexts, especially when facing selective pressures from the tumor microenvironment or cancer therapies [29,30,31,32,33]. While heterogeneity can confer certain benefits, such as increased proliferation and invasion, these features can also promote tumor progression, therapy resistance and disease recurrence [29,31,32]. There is an ongoing debate about the effectiveness of single-region biopsies in capturing the full complexity of HCC [32,33].

In this study, we introduce an enhanced HP algorithm optimized through training with additional multicenter data, with particular emphasis on its negative predictive value (NPV) and specificity. Our findings demonstrate that regardless of tumor heterogeneity—specifically the grade of tumor differentiation and histological type—independent sampling from the same nodule or different nodules within the same patient consistently provides reliable prognostic information.

## 2. Materials and Methods

### 2.1. Study Design and Population

This retrospective study analyzed HCC explant samples from patients who underwent LT alongside their corresponding clinical data. The study included three patient cohorts: Hospital Curry Cabral (Lisbon, Portugal, including patients transplanted between 1998 and 2012, *n* = 162 patients), Hospital Universitari I Politècnic La Fe (Valencia, Spain, 2014–2016, *n* = 70 patients) and Hospitais da Universidade de Coimbra, Unidade Local de Saúde de Coimbra (Coimbra, Portugal, 2014–2018, *n* = 24 patients). Two datasets were generated. Dataset 1 comprised all 232 patients from the Lisbon and Valencia cohorts, of which 162 had been part of the training of version 2.0 of the HP algorithm. This dataset was used for algorithm retraining and testing. Dataset 2 included the 24 patients from the Coimbra cohort, along with 22 patients from the Lisbon cohort from whom multiple samples had been collected. Dataset 2 was utilized to explore and characterize the impact of intra-nodule and intra-patient (inter-nodule) heterogeneity on the performance of the HP assay. Additionally, a subset of 141 samples from Dataset 1, for which data on AFP levels were available (termed HP AFP samples), was included for analysis where appropriate. The applied methodology is summarized in Figure 1.

Demographic data collected included patient gender, age and obesity status, as well as clinical information such as the number, size and volume of tumors. Additional patient details encompassed the MELD score, duration of the waiting list, survival time and recurrence date. Sample and data collection procedures received approval from the ethics committees at each participating center.

### 2.2. Sample Collection

This study utilized formalin-fixed paraffin-embedded (FFPE) HCC explants collected from patients who underwent LT. For samples included in Dataset 2, tumor tissue was macro-dissected either from at least two different regions within the same tumor nodule and/or from two distinct HCC nodules from the same patient. All HCC FFPE samples, from both Datasets 1 and 2, were sectioned (5 μm thick) using a microtome (Leica SM2010R Sliding Microtome, Leica Biosystems, Richmond, VA, USA) and mounted on glass slides. Tumor regions were evaluated by an experienced pathologist using a hematoxylin-eosin (HE)-stained consecutive section (3 μm thick). Larger nodules, which are more easily detected through pre-LT imaging, were most frequently selected for molecular analysis. The HCC regions analyzed were collected in a way that mimicked the sampling conditions of needle biopsies.

### 2.3. Sample Analysis and Selection

All HE-stained slides were evaluated by a pathologist to identify the relevant tumor areas for processing using the HP protocol. For Dataset 2, in addition to this evaluation, samples were analyzed by two independent certified pathologists. The tumor classification of histological patterns and grade of tumor differentiation was based on the latest World Health Organization classification (reviewed by [34]). Predominantly necrotic tumor areas were excluded from the analysis.

### 2.4. HepatoPredict Assay

The HP molecular protocol was applied to all selected tumor areas, as previously described [27,28]. Briefly, RNA was extracted from two consecutive 5 μm tissue sections per sample. An initial reverse transcription quantitative polymerase chain reaction (RT-qPCR) was performed to assess RNA quality and quantity, followed by a one-step RT-qPCR reaction to measure the expression levels of *DPT*, *CLU*, *CAPNS1* and *SPRY2* genes. Gene expression normalization was performed using the reference genes *RPL13A*, *GAPDH* and *TBP*, as detailed previously [27,28]. For the acquisition and analysis of qPCR data, we utilized QuantStudio Design & Analysis Software v1.5.1 (ThermoFisher, Waltham, MA, USA). The HP kit integrated the four gene expression signatures (molecular variables) with pre-LT clinical variables (number of tumors, diameter of the largest tumor, total tumor diameter) through a proprietary algorithm. This algorithm classified patients in terms of their predicted benefit from LT by categorizing them into three classes: Class II (benefit with high confidence), Class I (benefit with very high confidence, as a subset of Class II) and Class 0 (no predicted benefit from LT).

### 2.5. Performance Metrics

In this study we utilized several performance metrics to evaluate our model, including accuracy measures, such as sensitivity/recall, specificity, positive predictive value (PPV)/precision, NPV and overall accuracy. A brief description of these metrics and their respective formulas can be found in Appendix A.

### 2.6. HepatoPredict Algorithm Retraining and Performance Assessment

The retraining of the HP algorithm, first described by Pinto-Marques et al. [27] and subsequently improved [28], aimed at maximizing specificity, precision/PPV and sensitivity/recall. To evaluate the performance of our predictive model, we employed a cross-validation approach using repeated holdout validation. This method is similar to Leave-One-Out Cross-Validation (LOOCV), where the model is trained on all but one data point in each iteration by creating multiple training and testing splits to evaluate the model comprehensively. However, we partitioned the dataset into 70% training and 30% testing subsets instead of opting for single data point validation. This random partitioning was repeated 100 times to create independent train/test partitions, allowing for a thorough evaluation of the model’s performance across different data configurations. Each partition provided unique training and testing subsets, which helped us estimate the variability and generalizability of the model under various random splits. Given the imbalanced nature of our dataset, which had a higher proportion of patients who do not recur compared to those who do, we employed oversampling to the training data using the Adaptive Synthetic Sampling (ADASYN) technique [35]. After training, the performance of the retrained HP algorithm was assessed using the aggregated results from the testing subsets of each partition and the results were presented as mean with the respective standard deviation (SD).

### 2.7. Intra-Nodule and Intra-Patient (Inter-Nodule) Heterogeneity

To evaluate the performance of the HP algorithm concerning tumor heterogeneity, independent tumor samples were collected from at least two distinct regions within the same nodule and from distinct nodules in patients with multiple nodules. Two certified pathologists classified each sample based on tumor histological growth patterns and differentiation grades, which are a proxy for determining homogeneous versus heterogeneous classifications. Nodules or patients were classified as homogeneous if no differences were observed between samples, while those with variations were deemed heterogeneous. Intra-nodule heterogeneity was defined as the presence of two or more histological patterns or grades within the same nodule. Inter-nodule or intra-patient heterogeneity was identified when two nodules from the same patient exhibited differing classifications, even if only one sample showed a variation. If intra-nodule heterogeneity was detected in one nodule, inter-nodule heterogeneity was automatically assumed. Each sample underwent processing with the HP assay, which classified them individually as either Class II (indicating a predicted benefit from LT) or Class 0 (indicating no predicted benefit). The clinical outcome, specifically whether the patient experienced recurrence, was used to evaluate the accuracy of these predictions. The assessment of the HP assay focused on its concordance—whether the algorithm produces consistent results for different samples from the same nodule—and its ability to provide accurate prognosis in concordant samples. If the HP algorithm yielded discordant results across samples, no conclusion could be drawn regarding prediction accuracy and such results were automatically deemed incorrect.

### 2.8. Data Analysis and Visualization

The performance of the retrained HP algorithm was compared with published data from its previous version [28]. Additionally, it was directly compared with other criteria currently used in clinical practice for selecting HCC patients for LT. These criteria included the Milan criteria [8], the University of California, San Francisco (UCSF) criteria [36], AFP score [37], Metroticket 2.0 (MT2.0) [9], Argentinian score (ArgScore) [38], Warsaw criteria [39] and within all criteria (wALL) [40]. The calculations for these criteria are outlined in Appendix A. For the comparison involving AFP-including criteria (AFP score, MT2.0, ArgScore, Warsaw and wALL), only a subset of patients from Dataset 1 with available AFP data (HP AFP samples) was analyzed. The performance metrics for each criterion were evaluated across the same 100 testing subsets used to assess the HP algorithm. The results are presented as the mean and respective SD of the aggregated data (Figure 2). Data from the most representative testing subset (where all metrics were closest to the overall mean) were used for analyses requiring patient counts (Figure 3, Figure 4 and Appendix A). Recurrence-free survival (RFS) and overall survival (OS) curves and respective log-rank tests were generated using the ggsurvplot package on R studio version 2023.12.1+402. The follow-up time was calculated from the date of surgery (LT) to the last follow-up or the occurrence of an event (recurrence for RFS or death for OS).

## 3. Results

### 3.1. Demographic and Clinical Data Are Comparable Between Datasets

This study included patients diagnosed with HCC who underwent LT from three different centers in two countries. Among these patients, 94 had never undergone analysis through the HP assay [28]. Participants were included in Dataset 1, which was used for retraining the algorithm and in Dataset 2, which aided in investigating how HCC heterogeneity affected HP performance.

As detailed in Table 1, both datasets exhibited comparable demographic and clinical characteristics. Most patients were male (over 87%), with a median age of 57–58 years old and 26% were classified as obese. The typical waiting list time was approximately 2 months. Most patients were within Milan criteria (over 67%) with a total tumor volume of less than 115 cm^3^ (over 90%) which was reflected in an overall survival rate greater than 60% and a recurrence rate lower than 20% at 5 years (Table 1). Notable differences between the datasets included the median number of nodules and the slightly larger total tumor diameter and volume (Table 1). These discrepancies arise from the multi-nodular nature of the tumors in Dataset 2, which were intentionally selected to facilitate the examination of inter-nodular heterogeneity and its impact on HP performance.

### 3.2. The Retrained HP Algorithm Outperforms the Current Clinically Used Criteria

Since HP is a machine learning-based tool, it is essential to continuously train it with new representative data from the population of interest, especially in situations involving imbalanced data. In this context, a dataset of 232 patients (Dataset 1) was utilized for retraining, which included a new cohort from a different center and country (Valencia cohort, *n* = 70). The correlations between HP classification and patient outcomes (RFS and OS) showed no significant differences across either cohort in Dataset 1 (Appendix A). Compared to its earlier version (as published [28]), the retrained HP algorithm demonstrated improvements across all performance metrics (Appendix A), particularly in sensitivity/recall (increased from 0.91 to 0.96), NPV (increased from 0.56 to 0.77) and accuracy (increased from 0.79 to 0.85).

The performance of the retrained HP algorithm in the testing subset(s) was directly compared with several commonly used clinical criteria for selecting patients for LT, including Milan [8], UCSF [36], AFP score [37], MT2.0 [9], ArgScore [38], Warsaw [39] and wALL [40]. This comparison focused on various accuracy measures (Figure 2 and Appendix A), including the ability to correctly predict recurrence or no recurrence (Figure 3), RFS (Figure 4) and OS (Appendix A). For each criterion, patients were classified based on whether they were predicted to benefit from LT (termed “within criteria”, “included”, “IN” or Class II/I for HP) or not (“outside criteria”, “excluded”, “OUT” or Class 0 for HP).
Figure 2Performance metrics/accuracy measures of the retrained HepatoPredict algorithm and other currently used clinical criteria in the testing subsets. Sensitivity (Sen), positive predictive value (PPV), specificity (Spe), negative predictive value (NPV) and accuracy (Acc) were measured. Data is represented as mean. The retrained HepatoPredict (HP) was compared with Milan criteria (Milan) and the University of California, San Francisco (UCSF) criteria (*n* = 69), whereas the HP AFP samples subset was compared with AFP-based criteria such as AFP score, metroticket 2.0 (MT2.0), Argentinian score (ArgScore), Warsaw criteria (Warsaw) and within all criteria (wALL) (*n* = 42). rHP Class I is a subset of rHP Class II.
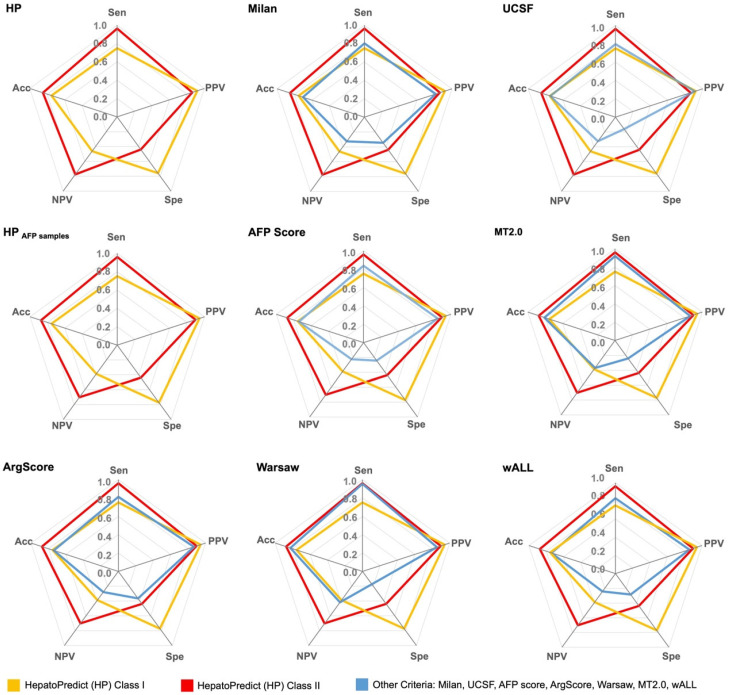

Figure 3Ability to correctly predict whether patients will or will not recur according to different criteria. Outcomes—no recurrence and recurrence—from the most representative testing subset (*n* = 70) and number and percentage of correct predictions according to the retrained HP algorithm (HP), Milan criteria (Milan) and University of California, San Francisco criteria (UCSF) (**A**). The same analysis was performed for patients from the HP AFP samples subset (*n* = 42) (**B**).
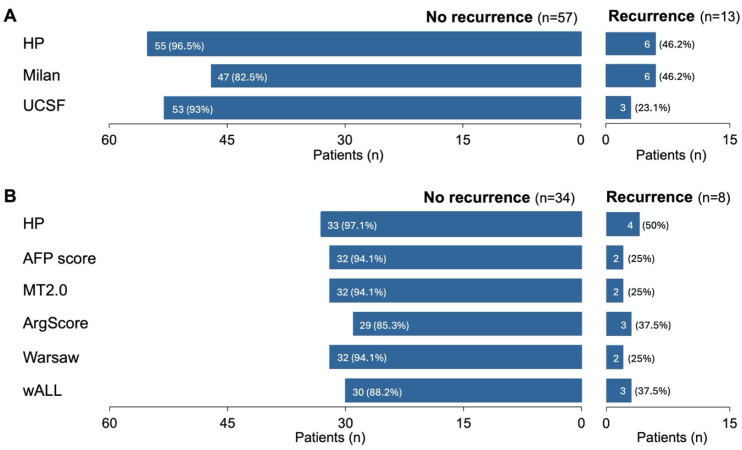

Figure 4Recurrence-free survival of patients according to different criteria. The recurrence-free survival (RFS) curves for the most representative testing subset are illustrated. The retrained HepatoPredict (HP) algorithm, Classes I, II and 0 (**A**), was compared with the Milan criteria (**B**) and the University of California, San Francisco (UCSF) (**C**) criteria, *n* = 68. Additionally, RFS was also calculated for the subset of patients with AFP values within the different HP classes (**D**). This was further compared with AFP-based criteria, including the AFP score (**E**), Metroticket 2.0 (MT2.0) (**F**), Argentinian score (ArgScore) (**G**), Warsaw criteria (**H**) and within all criteria (wALL) (**I**), *n* = 44. For each criterion, patients were categorized as eligible (IN) and non-eligible (OUT) for LT. HP Class I is a subset of HP Class II. The log-rank test, based on RFS analysis (**A**), showed significant differences between HP Class I vs. Class 0 (χ^2^ = 20.42, Bonferroni-adjusted *p* < 0.001) and HP Class II vs. Class 0 (χ^2^ = 12.18, Bonferroni-adjusted *p* < 0.01), but not between HP Class I and Class II (χ^2^ = 1.08, Bonferroni-adjusted *p* = 0.90). Additionally, for the cohort with AFP values (**D**), the long-rank results also showed significant differences between HP Class I vs. Class 0 (χ^2^ = 8.90, Bonferroni-adjusted *p* < 0.01) and HP Class II vs. Class 0 (χ^2^ = 7.16, Bonferroni-adjusted *p* = 0.022), but not between HP Class I and Class II (χ^2^ = 0.26, Bonferroni-adjusted *p* = 1.00). In contrast, all other tested criteria did not show significant differences between IN and OUT groups using the same log-rank test, highlighting the superior discriminatory power of the HP classification system.
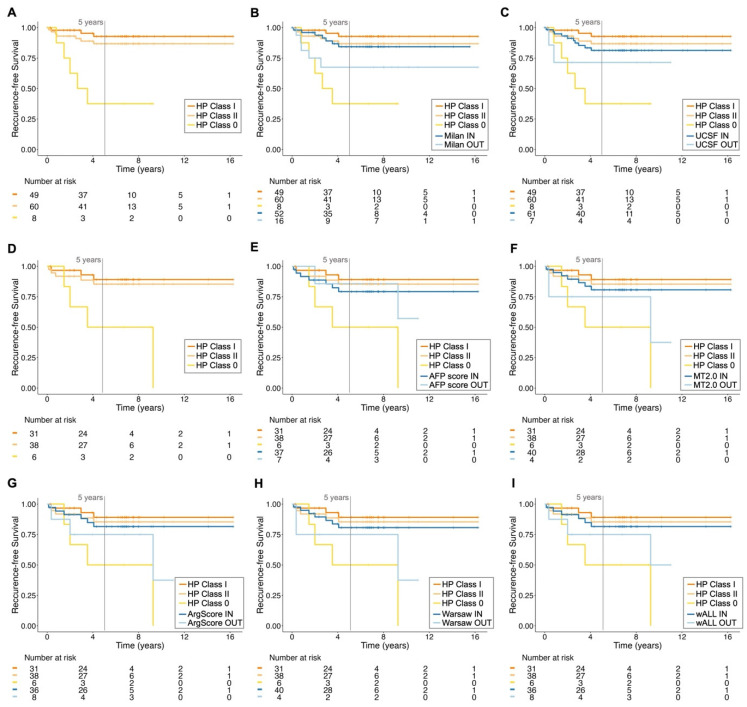



Compared to other criteria, including AFP-based ones, the HP algorithm demonstrated the best performance in accurately classifying patients who would benefit from LT and those who would not. Overall, it achieved an accuracy of 0.85, with 0.89 for the AFP samples subset. HP exhibited a high sensitivity/recall of 0.96 for the general set (0.97 for the AFP subset) and a PPV/precision of 0.86 (0.91 for the AFP subset), indicating HP effectiveness in correctly identifying patients who are not likely to recur (Figure 2, Appendix A). Moreover, the HP algorithm had a better specificity of 0.44 (0.47 for the AFP subset) and particularly a high NPV of 0.77 (0.74 for the AFP subset), which reflects its capability in correctly identifying patients who are likely to recur (Figure 2, Appendix A). Within the HP classification, HP Class I (a subset of Class II) displayed the highest specificity of 0.76 (0.81 for the AFP subset) and PPV of 0.92 (0.95 for the AFP subset). However, it had a lower sensitivity and NPV (Figure 2, Appendix A), which is in alignment with its intended purpose of selecting patients with very high confidence in their predicted benefit from LT.

In line with the previous results, HP also excelled at correctly predicting whether patients would recur or not (Figure 3). When considering both the most representative testing subset and the AFP samples subset, HP outperformed other criteria, correctly predicting no recurrence in 96.5% and 97.1% of cases, respectively (Figure 3). For patients who did recur, HP was non-inferior to or better than other criteria, making correct predictions in 46.2% and 50% of these cases (Figure 3). However, these values may be partly influenced by the limited number of patients who recurred—approximately 19% in this testing subset—which was consistent with the overall dataset (Table 1).

The median follow-up period for patients in Dataset 1 was 6.6 years, with a maximum of 16 years (Table 1). Thus, we analyzed RFS (Figure 4) and OS (Appendix A) over 16 years for each HP classification and compared them with the other criteria. Consistent with previous findings [41,42,43], most HCC recurrences in our dataset occurred in the first two years post-LT, which is reflected in the similar RFS values at 5 and 16 years for all criteria, suggesting the long-term validity of HP predictions (Figure 4). HP demonstrated the greatest and only statistically significant discrimination between the IN (Class II and Class I) and OUT (Class 0) categories. At 5 years, more than 80% of patients deemed eligible for LT had not recurred, whereas at least 50% of the noneligible patients had recurred (Figure 4). This reinforces the notion of a good NPV (Figure 2, Appendix A), despite the limited number of patients in this context. Moreover, both IN HP classes showed comparable RFS to all other IN populations using different criteria, particularly in the long-term (Figure 4).

Regarding OS, no significant differences were found, but a trend was observed for HP Class I patients to have the highest survival rates (~75% at 5 years, Appendix A), while those in HP Class II demonstrated comparable outcomes to the IN classification of all other criteria (>60%, Appendix A). On the other hand, patients in HP Class 0 tended to have the lowest OS among the OUT classification of all other criteria (<40%, Appendix A).

### 3.3. The HepatoPredict Tool Demonstrates Strong Performance Despite Intra-Nodule and Intra-Patient Heterogeneity

This study aimed to assess how tumor heterogeneity influenced the HP algorithm’s effectiveness. Dataset 2 included patients with at least two samples from distinct tumor regions per nodule, enabling the evaluation of both intra-nodule and inter-nodule/intra-patient heterogeneity in multi-nodular tumors. A total of 158 independent tumor samples from 77 nodules were collected from 46 patients in Dataset 2 (Appendix A). The overall accuracy of HP in this dataset (0.84, Figure 5A), measured across the 158 independent samples, was consistent with its performance in Dataset 1 (0.85, Figure 2, Appendix A).

Histopathological evaluation classified nodules/patients as homogeneous (no differences in the histopathological classification between samples) or heterogeneous (differences present). Most samples exhibited a trabecular growth pattern (65%), followed by solid (15%), pseudoglandular (5%) and macrotrabecular types (5%). Six percent displayed mixed patterns, combining trabecular with another type, while 4% were classified as the steatohepatitic subtype. In terms of tumor differentiation, 30% were well differentiated (G1), 68% moderately differentiated (G2) and 1% poorly differentiated (G3).

The impact of intra-nodule heterogeneity on HP performance was evaluated in 77 nodules and 46 patients (Figure 5, Appendix A). Overall, 83.1% of analyzed nodules were concordant, meaning they had the same HP classification despite different sample collections (Figure 5A). In these concordant nodules, HP correctly predicted patient prognosis in 90.6% of cases (Figure 5A). For patients, 80.4% showed concordant intra-nodule HP classifications, with 89.2% having correct predictions (Figure 5B).

Considering the clinical practice of performing a biopsy on the largest visible nodule, we examined the impact of intra-nodule heterogeneity on HP performance between the largest nodule and other smaller nodules (Figure 5C). No meaningful differences were found in the percentage of concordant nodules (82.6% for the largest vs. 83.9% for others) or in the correct predictions (92.1% for the largest vs. 88.5% for others).

Differences were more pronounced when comparing HP performance in homogeneous nodules vs. heterogeneous ones (Appendix A). Heterogeneous nodules showed less concordance (75% vs. 82.4%) and fewer correct HP predictions (77.7% vs. 92.9%) compared to homogeneous nodules (Appendix A). However, these results may be influenced by the limited number of heterogeneous nodules (only 12).

The impact of inter-nodule (intra-patient) heterogeneity on HP performance was assessed in 28 multi-nodular patients (Appendix A). HP performance was compared between patients with inter-nodule heterogeneity and those with similar histopathology. HP concordance across different nodules was around 68%, regardless of homogeneity (Appendix A). In concordant patients, HP accurately predicted prognosis 100% of the time (Appendix A).

## 4. Discussion

The global incidence of HCC is expected to rise, leading to an increased demand for LT, which remains the most effective curative treatment for this cancer [4]. Ensuring fair and accurate patient selection for LT is crucial, as inadequate selection poses significant clinical challenges and raises important ethical concerns [23,25,26]. To tackle this issue, we developed and refined the HP tool, which integrates molecular data from tumor biopsies with clinical variables through a machine learning approach to predict the benefits of LT for HCC patients [27,28].

HP has previously demonstrated superior accuracy compared to various selection criteria [27,28]. In this study, the improved HP algorithm, trained with a two-site and representative dataset, once again outperformed commonly used clinical criteria for selecting patients for LT. HP exhibited the highest sensitivity and PPV/precision, effectively identifying patients who would benefit from LT by accurately predicting non-recurrence outcomes. Additionally, HP demonstrated higher specificity and improved NPV, while maintaining comparable performance in identifying recurrence cases. Notably, HP class I, a more stringent stratification option representing a subset of class II, offers higher specificity and PPV, identifying a group of patients where the recurrence rate is drastically decreased (8%), thereby demonstrating its value as a tool for when organ availability is scarce.

Moreover, the performance metrics for all criteria, except for specificity (which was generally lower in our study), closely aligned with those reported in a recent systematic review of 14 different LT selection criteria [23]. AFP-based criteria (AFP score [37], MT2.0 [9], ArgScore [38], Warsaw [39] and wALL [40]) performed only slightly worse than HP regarding sensitivity and PPV. This was to be expected, since integrating biological factors like AFP or DCP improves stratification accuracy compared to purely morphological criteria like Milan [44]. However, the NPV—indicating the probability of correctly identifying poor prognosis—remained below 50% for all criteria except for the HP tool, which achieved an NPV of 77%. While higher PPV and sensitivity allow for broader patient selection, the low NPV in other criteria suggests many patients with good prognosis are still being incorrectly excluded from receiving LT, thus denying them the best standard of care and raising ethical concerns. Previous studies have highlighted the need for improved NPV in LT selection criteria [23]. The retrained HP algorithm improved all metrics, particularly NPV, resulting in a balanced performance, indicating its potential to address equity issues in LT selection for HCC.

The improved performance of HP is likely due to two main factors. First, as a machine learning tool, HP aligns with the hypothesis proposed by Lai and colleagues that combining biological data with artificial intelligence approaches can enhance LT selection accuracy [44]. Second, HP incorporates molecular variables from tumoral tissue, offering a more direct assessment of tumor biology than biomarkers like AFP and DCP, which have limited sensitivity and specificity [18,19,20].

The HP IN vs. OUT classes were also significantly associated with long-term RFS outcome. Over 80% of patients predicted by HP to benefit from LT (class II and class I) did not experience recurrence, while at least 50% of those predicted not to benefit did experience recurrence. Compared to patients deemed eligible by other criteria, those in classes II and I had equivalent long-term RFS. Importantly, HP exhibited the greatest and only statistically significant ability to differentiate between patients with and without predicted benefits for long-term RFS, highlighting the superior discriminatory power of HP in this context. No significant differences were found for OS for any of the criteria analyzed, suggesting that factors other than recurrence affect patient survival.

The clinical application of the HP tool requires an ultrasound-guided biopsy of a visible tumor. A limitation of biopsies is that the small tissue samples may not accurately reflect the tumor’s morphological, phenotypic and molecular heterogeneity [45]. Therefore, it is essential to consider whether the HP approach, reliant on biopsies, introduces sampling bias in the context of HCC heterogeneity.

Our dataset revealed 26% of intra-nodule heterogeneity and 79% of inter-nodule (intra-patient) heterogeneity, consistent with previous reports [46,47]. The degree of intra-tumor heterogeneity varies significantly among HCC patients [48], identified in 12–66% of cases, as reviewed in [33]. Regardless of intra-nodule heterogeneity, HP demonstrated high concordance (>80%) and a strong ability to correctly determine patient prognosis in nodules with intra-nodule HP concordance (>89%). In practice, our results indicate that sampling a nodule twice yields at least an 80% chance of obtaining the same HP classification. Moreover, results remained consistent whether analyzing the largest nodule or other nodules, indicating that HP performs well even when the largest nodule is unavailable for sampling. We observed a slight decrease in concordance (5–13%) when focusing on patients with intra-nodule heterogeneity or multiple nodules (regardless of inter-nodule heterogeneity). However, these findings can be partly attributed to a small sample size and, more significantly, to our stringent classification of all discordant nodules as incorrect.

We provide evidence that even if a single-region biopsy does not fully capture the complexity of heterogeneous tumors, it remains representative for HP classification, regardless of the biopsy site within the nodule. Overall, HP exhibits good prognostic power independent of nodule heterogeneity, particularly regarding histological type and differentiation grade. This indicates that the molecular variables considered by HP effectively capture the biological variation within tumors, even with a single biopsy and even when that biopsy is not taken from the largest nodule. Additionally, our data indicate that if the largest nodule is inaccessible, biopsies from other nodules can still provide valuable insights into tumor behavior. This is crucial, as biopsy procedures should minimize the impact on the liver and patient as much as possible [34].

Our findings also reinforce the value of incorporating pathological examinations into the HCC patient journey and support the potential of HP in real-world applications. Historically, most HCC management guidelines have indicated that imaging techniques alone suffice for diagnosis, reserving biopsies for non-cirrhotic liver or ambiguous imaging results [47,48]. Advances in imaging and concerns about biopsy procedures (needle tract seeding, sampling errors, small risk of morbidity) have contributed to this perspective [47]. However, the latest HCC management EASL guidelines (2018) state that “it is now widely accepted that the potential risks, bleeding and needle track seeding, are infrequent, manageable and do not affect the course of the disease or overall survival. In general, they should not be seen as a reason to abstain from diagnostic liver biopsy” [49]. The growing evidence on the information provided by tissue samples is likely to modify the current risk/benefit ratio of biopsies [45]. Pathologists have identified a diverse range of morphology-associated histological features linked to different HCC subtypes, which correlate with specific tumorigenic gene mutations and transcriptomic profiles [50,51,52,53]. For instance, *CTNNB1* mutations are associated with well-differentiated tumors exhibiting microtrabecular and pseudoglandular patterns, which correlate with better prognosis and less aggressive subgroups [53]. Contrarily, *TP53* mutations are found in poorly differentiated cancers with macrotrabecular and solid patterns [52].

This study has several limitations. Due to the scarcity of biopsies associated with LT in retrospective cohorts, we used HCC surgical explants as a proxy, collecting small tumor areas (10 × 2 mm) to mimic a typical liver biopsy [54]. While this approach may not perfectly replicate biopsy conditions, previous studies have demonstrated good concordance between tumor biopsies and larger specimens [47]. However, it is important to consider the time–LT variable when extrapolating these results, particularly in regions with longer waiting lists. Prolonged waiting times, downstaging or bridging therapies can all influence tumor biology [55,56,57]. Another limitation is the imbalance in our dataset, with fewer than 20% of patients showing tumor recurrence, which may lead to overpredicting the major class. Nonetheless, our cohorts reflect real-world recurrence rates of 8–20%, as reviewed in [58] and we employed the appropriate methodology to account for this imbalance when retraining the algorithm. The next step is to address HepatoPredict performance in a totally independent test set. With this in mind, additional retrospective cohorts are underway, as well as a prospective study (NCT04499833), currently recruiting participants. Although small, the Coimbra cohort serves as an independent validation set, comprising approximately half of the samples in Dataset 2 (24 of 46 patients). This suggests that HP performance remains robust, demonstrating an accuracy of 84%, consistent with the metrics from Dataset 1.

## 5. Conclusions

The HP tool outperformed commonly used clinical criteria for selecting patients for LT, particularly excelling in NPV, which addresses a critical gap in current selection strategies. Importantly, HP’s prognostic power remained largely unaffected by nodule heterogeneity. This suggests that the molecular variables it analyzes effectively capture the biological diversity within tumors, even when biopsies are taken from nodules other than the largest. Overall, the HP tool represents a significant advancement in achieving equitable and accurate patient selection for LT, allowing more patients to access the best standard of care while optimizing organ allocation and improving outcomes.

## Figures and Tables

**Figure 1 cancers-17-00500-f001:**
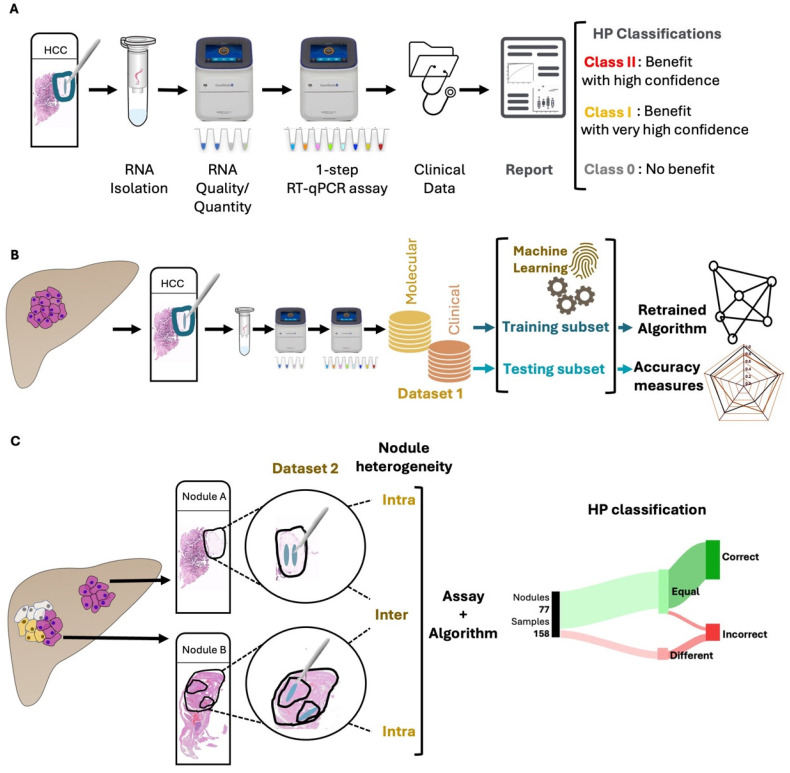
Methodology Workflow. All collected samples were tested, individually, as previously described [27,28] using the HepatoPredict molecular assay and algorithm (**A**). In Dataset 1, RNAs were isolated from FFPE liver explant HCC nodules and analyzed with the HepatoPredict molecular assay. The molecular and clinical variables from Dataset 1 samples were used to retrain the HepatoPredict algorithm using machine learning approaches and to calculate accuracy measures for the new version (**B**). In Dataset 2, at least two representative samples, mimicking a tissue biopsy, were collected from both homogeneous and heterogeneous nodules, based on a pathologist’s analysis of a hematoxylin-eosin-stained slide. The performance of HepatoPredict was then assessed in samples isolated from the same nodule (to investigate the impact of intra-nodule heterogeneity), as well as in samples isolated from distinct nodules within the same patient (to explore the effects of inter-nodule heterogeneity) (**C**).

**Figure 5 cancers-17-00500-f005:**
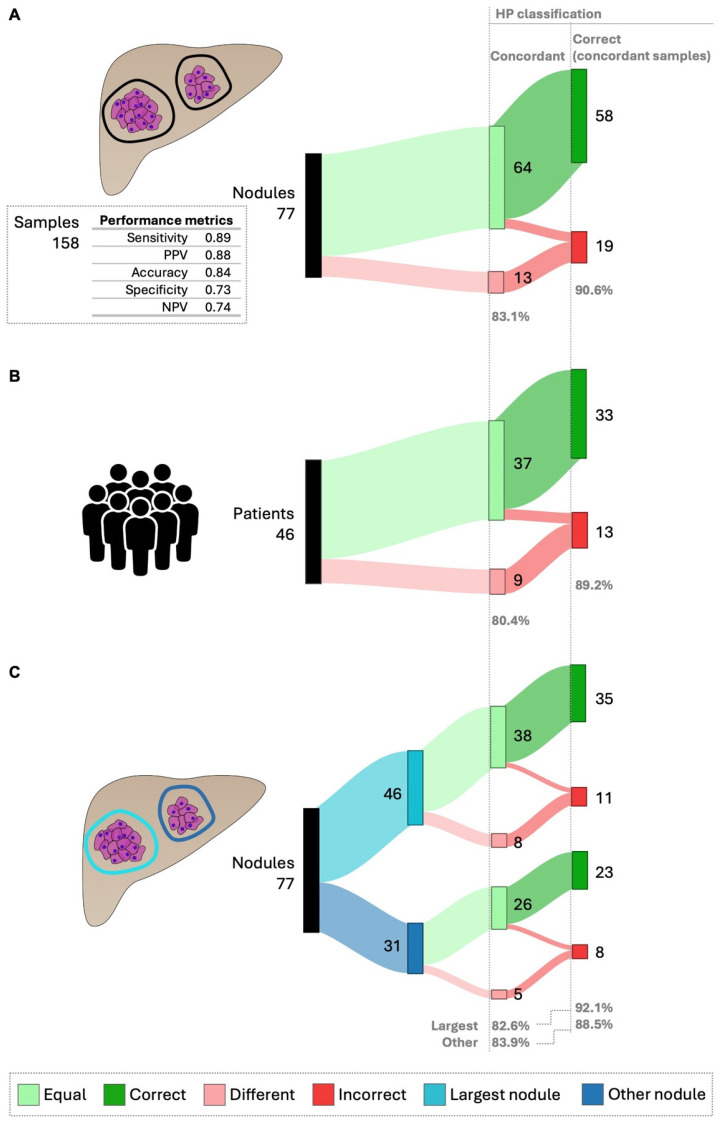
HepatoPredict performance in the context of intra-nodule heterogeneity—concordance and correct prediction in concordant samples. To evaluate the impact of intra-nodule heterogeneity on HP performance, at least two samples of each nodule were collected and characterized regarding the concordance of their HP assay results, specifically, whether these samples from the same nodule received the same HP classification. The performance of HP, defined as its ability to produce correct prognoses, was evaluated using only the HP-concordant samples. HP performance metrics for Dataset 2, which includes data from 158 samples, are shown (**A**). This analysis was further performed for individual patients (**B**) and the results from the largest nodules were also compared with those from other nodules (**C**). The number (N) and percentage (%) are indicated for nodule/patients showing concordant vs. different HP results and for concordant samples. Any instance where the HP algorithm produced discordant results was automatically classified as incorrect.

**Table 1 cancers-17-00500-t001:** Demographic and clinical characteristics of the patients included in each dataset.

	Dataset 1 (*n* = 232)	Dataset 2 (*n* = 46)
**Recipient characteristics**		
Male gender, N (%)	203 (87.5%)	44 (95.7)
Age, years, median (IQR)	57 (13)	58 (14)
Obesity, N (%)	61 (26.3)	12 (26.1)
MELD score, median (IQR)	11 (5.5)	12.4 (5)
Waiting list, months, median (IQR)	2.2 (4.1)	1.8 (2)
**Tumor-related factors**		
Nº of nodules, median (IQR), range	1 (1), 1–4	2 (1), 1–6
Size of largest nodule, median (IQR), range	2.9 (1.8), 0.7–8.7	3.0 (1.6), 1.0–8.2
Total tumor diameter (cm), median (IQR), range	3.4 (2.5), 0.7–12.5	4.45 (2.3), 1–12.5
Total tumor volume (cm^3^), median (IQR), range	14.1 (28.3), 0.2–344.8	19.5 (28.9), 0.5–291.4
Within Milan criteria, N (%)	179 (77.2)	31 (67.4)
Total tumor volume ≤ 115 cm^3^, N (%)	219 (94.4)	42 (91.3)
**Survival data**		
Patients alive at 5 years, N (%)	151 (65.1)	29 (63.0)
Recurrence at 5 years, N (%)	43 (18.5)	9 (19.6)
**Follow-up (years)**		
Follow-up, median (IQR), max. range	6.6 (5.9), 16.3	5.0 (4.3), 15.9

N—number of samples, IQR—inter-quartile range, MELD—model for end-stage liver disease.

## Data Availability

The datasets used and/or analyzed during the current study are available from the corresponding author upon reasonable request.

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
