# Peer review of "HepatoPredict Accurately Selects Hepatocellular Carcinoma Patients for Liver Transplantation Regardless of Tumor Heterogeneity"

_cancers, 2025, doi:10.3390/cancers17030500_

Round 1

Reviewer 1 Report

Comments and Suggestions for Authors

This study by Andrade et al. addresses a currently important topic. It is about improving the correct identification of LT candidates that should have a lower risk of HCC recurrence than the current 20% generated from numerous scoring systems. Internationally, the number and size of nodes with or without AFP values are taken into account differently.
Andrade et al. go into more depth and investigate the heterogeneity of the tumors in a large cohort of HCC patients from three transplant centers using gene expression analysis, liver bipsies, clinical data and AI algorithm formation, among other things, to create a new model with better performance than the previous scoring systems. 

Overall, it is a well-written paper, albeit somewhat overloaded. The introduction could be a bit shorter.

One important point is not considered or not stated.
The extent of immunosuppression is not shown. The groups to be compared should have a comparable IS load, as this can significantly alter recurrence and survival afterwards. Perhaps the authors could address this aspect.

Author Response

We are grateful to the reviewer for the positive appreciation of our work and for the helpful comments.

While we agree that the extent of immunosuppression can alter recurrence and survival outcomes, we chose not to address this in our work for several reasons. Although not shown here, upon development of HepatoPredict an analysis was conducted to assess which clinical variables influenced prognosis (disease recurrence) in patients that underwent liver transplantation. Different immunosuppression schemes did not have a significant influence, i.e., this variable showed no predictive power regarding disease recurrence. This is in line with the results of Figure S2, which show that there are no differences in the RFS curves of each HepatoPredict class when comparing the two centers of Dataset 1 (Portugal and Spain). These centers likely have different immunosuppression schemes, more so because patient collection occurred in different time intervals (1998-2012 in Portugal and 2012-2014 in Spain). RFS curves do not show differences between the two centers favoring the reasoning that, even in the presence of immunosuppression variability between patients and centers the HepatoPredict signature is capable of capturing the outcome. OS curves do show some differences, likely due to other variables, the most relevant of which is potentially the time spent on the waiting list but this was not further addressed in the present work. Taking the above-described facts into account, we chose not to address immunosuppression load in the present work.

Reviewer 2 Report

Comments and Suggestions for Authors

The authors found that HepatoPredict accurately selects hepatocellular carcinoma for patients for liver transplantation regardless of tumor heterogeneity. There are still some problems in the manuscript and below are my comments:

1.     Milan or UCSF criteria are important in selecting pt for liver transplant for HCC. Tumor heterogeneity did not influence selecting pts for liver transplant and liver biopsy was not usually performed before liver transplant.

2.     The differentiation of grade I and II HepatoPredict is not large.

3.     Overall survival and recurrence free survival are both important. In figure 4 , S2 and S3, the P value was not given and the superiority of Hepatopredict was not shown.

4.     Could the HepatoPredict replace the Milan or UCSF criteria for selecting pt for liver transplant?

5.     Will the Milan or UCSF criteria plus the HepatoPredict enhance overall survival ? ( the manuscript should present this question ) And if pts beyond the Hepatopredict but within the Milan or UCSF criteria, will the authors be against liver transplant for these pts?

6.     The etiology of all pts with advanced liver disease should be told and compared ( eg, MASLD, HBV or HCV related diseases)

7.     The authors concluded “These findings support the usage of HP as a valuable tool for optimizing LT candidate selection, promoting fair organ allocation, and enhancing patient outcomes through integrated analysis of molecular and clinical data.” . How did the result of the study confirm their conclusions?

Author Response

1. We agree that Milan and UCSF criteria still have an important role in selecting patients for liver transplantation (LT), being widely used and having a well-known performance. However, as we mention in our Introduction (page 2, paragraphs 3-4) and as we also found in this study (Figure 2, Table S3), these criteria display low negative predictive value (NPV), thus wrongly excluding patients with good prognosis who could benefit from the procedure. HepatoPredict can help to accurately select a higher number of patients, thus contributing to a more equitable organ allocation.

We are also in agreement that, so far, no patient selection criteria for LT rely on biopsy, and we expand on this subject in our Discussion. We consider that use of tumoral tissue (a direct measure of tumor biology) is  the main contribution for  HepatoPredict’s overall performance (page 14, paragraph 4 of the Discussion). Furthermore, the growing evidence on the information provided by tissue samples is enforcing the importance of modifying the current perception of the risk/benefit ratio of a biopsy (page 15, paragraph 4). However, taking a biopsy implies considering the likelihood of tumor heterogeneity, and this is one of the main reasons why our analysis of heterogeneity contribution was performed in the present study.

Altogether, we consider that these topics are discussed at length in our study.

2. We acknowledge the reviewer for addressing this topic. However, we are unclear on what the reviewer meant here – the difference between HepatoPredict class I and class II? Assuming this is the case, we reinforce throughout the manuscript that HepatoPredict class I is a more stringent subset of class II where patients are predicted to benefit from LT with higher confidence. This results in a higher positive predictive value (PPV) and specificity but lower NPV and sensitivity (Figure 2, Table S3). These differences are also reflected in the RFS and OS curves (Figure 4, Figure S3). Thus, as we propose in our Discussion (page 14, paragraph 2 of the Discussion), the HepatoPredict classes might be useful in different clinical contexts: class I may be a valuable tool when organ availability is scarce, whereas class II can be used when organs are more readily available to select a higher number of patients accurately.

3. We appreciate the reviewer's input and have now conducted log-rank statistical analysis, the results of which have been added to the legends of these figures. We have also added this to the Methods section (page 6 section 2.8) and adjusted our discourse in the Results and Discussion sections when appropriate (pages 7 and 11 section 3.2 and page 14 final paragraph). By conducting this analysis, we statistically confirmed the superior discriminatory power of the HP classification system vs. the other criteria, and thus we thank the reviewer for the pertinent comment. The same statistical significance was not found in the overall survival data for any of the criteria, which suggests that factors other than recurrence affect patient survival, a point which we added to our Discussion (page 14 last sentence page 15 first sentence).

4. As mentioned in a previous response, we agree that Milan and UCSF criteria have an important role in selecting patients for LT, being widely used and having a well-known performance. However, as stated in our manuscript and in a recent meta-analysis currently submitted for revision (doi: 10.1101/2024.12.16.24319098), these criteria display low NPV, thus wrongly excluding patients with good prognosis who could benefit from the procedure. HepatoPredict can help select a higher number of patients accurately, thus contributing to a more equitable organ allocation. Given the enhanced performance of HepatoPredict and upon further validation on more independent cohorts and prospective studies, which we are currently addressing, we would support the replacement of the criteria.

5. Regarding the usage of Milan/UCSF criteria plus HepatoPredict we did not pose this question for two main reasons. Firstly, Milan criteria are very restrictive and still result in recurrence rates up to 20%, whereas UCSF criteria include more patients with even higher recurrence rates. Using HepatoPredict we demonstrate that we can select more patients with better accuracy compared to both Milan and UCSF criteria. Secondly, the HepatoPredict algorithm includes morphological clinical variables (as detailed in the Introduction and Methods sections), meaning that a joint usage with morphological criteria like Milan and UCSF is not expected to add value.

Additionally, HepatoPredict was developed having the risk of recurrence as the output variable. We have now shown statistically significant differences for the RFS outcome between HP IN and OUT classes. On the other hand, analysis of the overall survival data did not find statistically significant differences for any of the criteria. These data suggests that while HepatoPredict and other criteria address (with different levels of success) RFS, unsurprisingly, they do not correlate with overall survival data.

Regarding the issue of patients beyond HepatoPredict but within the Milan/UCSF criteria, as stated in the previous question, given the enhanced performance of HepatoPredict and upon further validation on more independent cohorts and prospective studies, which we are currently addressing, we would support the result of HepatoPredict over the other criteria.

6. Thank you for pointing the etiology of HCC which is indeed an important factor in HCC progression. For the sake of not having a very extensive manuscript with data not contributing to the current HepatoPredict algorithm, during the development of HepatoPredict an analysis was conducted to assess which clinical variables influenced prognosis (disease recurrence) in patients who underwent liver transplantation. Different etiologies did not have a significant influence, i.e., this variable showed no predictive power regarding disease recurrence.

7. We appreciate the reviewer’s comment and the opportunity to clarify this. We based this conclusion on our results showing that: 1) HP outperforms the most used clinical criteria (Supp. Table S3), particularly regarding the negative predictive value metric, allowing the selection of a higher number of patients for LT more accurately; 2) HepatoPredict had the most balanced performance regarding accuracy measures, which is in line with a recently reported unmet need for LT selection tools which take into account the perspectives of different stakeholders, i.e., patients, physicians, payers and organ allocation organisms (doi: 10.1101/2024.12.16.24319098); 3) HepatoPredict demonstrated the greatest and only statistically significant discrimination between the IN (Class II and Class I) and OUT (Class 0) categories compared to the other criteria (Figure 4); 4) Despite being a tissue-based approach, the prognostic accuracy of HepatoPredict remains largely unaffected by intra-nodule and intra-patient heterogeneity, indicating its robustness even when biopsies were taken from smaller or non-dominant nodules (Figure 4).

Thus, overall, HepatoPredict is potentially a significant advancement in achieving equitable and accurate patient selection for LT, allowing more patients to access the best standard of care while optimizing organ allocation and improving outcomes. Further validation is on the way to further sustain these advancements.